# The use of virtual exhibition to promote exhibitors' pro-environmental behavior: The case study of Zhejiang Yiwu International Intelligent Manufacturing Equipment Expo

Qing Xia[1], Shan Wang[1]*, Jose Weng Chou Wong[2]*

1 School of Management, Guangdong University of Education, Guangzhou, China, 2 Faculty of Hospitality and Tourism Management, Macau University of Science and Technology, Taipa, Macau

* wangsh89@outlook.com (SW); wcwong@must.edu.mo (JWCW)

**Data Availability Statement:** All relevant data are within the paper and its Supporting information files.

## Abstract

This study investigates the role of virtual exhibition attributes (navigation, ubiquity, vividness, interactivity, visualization) in creating positive perceived green performance and satisfaction of exhibitors, thus benefiting the exhibitors' sustainable behaviors of Eco-exhibition. Two studies were conducted to verify the proposed hypotheses. In Study 1, 417 samples were collected from 2021 ME-Expo of China to test the model. In Study 2, the follow-up interviews were conducted with 18 participants to validate the quantitative results and gain deeper insights. The results of Study 1 indicate that adopting virtual exhibitions is critical in predicting exhibitors' perceived performance and satisfaction, which in turn, influences their pro-environmental behavior. The results of Study 2 confirmed above mentioned relationship, and interviewees indicate that the emergence of virtual exhibitions should be a long-term strategy for sustainable development in the exhibition industry.

## 1. Introduction

Recently, virtual meetings and exhibitions have been widely used and grown rapidly in the MICE industry, especially after the crisis of COVID-19 [1,2]. Virtual exhibitions enable users to participate in the exhibitions without the restrictions of space and time. Extant literature has identified that virtual exhibition provides a platform for visitors to experience an exhibition without leaving their current location [3]. For instance, after COVID-19, some studies showed that more than 60% of cultural museums worldwide have added the functions of online virtual exhibition, and Google has also partnered with more than 2,000 institutes in over 80 countries to provide virtual exhibition services [4]. While many current studies have explored the concept of virtual exhibition [5], or discussed how this technology can be used in different fields such as education [6], to our best understanding, there are no studies address what general attributes of virtual exhibition can help attract participants to join.

The construction materials often cause environmental pollution in destinations [7]. In the exhibition industry, organizers and exhibitors aim to create an attractive stand, thus overusing numerous wood-based or plastic-based materials [8]. Such high volumes of waste from

**Funding:** The author(s) received no specific funding for this work.

**Competing interests:** The authors have declared that no competing interests exist.

exhibitions have gained much attention from authorities and these issues are waiting to be solved. Past studies have posited that the use of web-based technology (etc. virtual tours) helps the firm to reduce costs, conserve the environment, and build a positive image [9]. The positive impacts of adopting web-based technology have been recognized by tourism industries [10]. Virtual technologies can help promote sustainable consumption by reducing unnecessary greenhouse gas emissions from transportation and enhance 'virtual accessibility' especially for the elderly and disabled visitors with limited mobility [11,12]. Therefore, the exhibition industry may benefit from the application of virtual technologies, to reduce overusing construction materials and environmental pollution.

In current studies of virtual exhibition, most of the current studies in the field of virtual exhibition focused on their convenience, and whether participants intend to use it continuously or not. For instance, Wu, Jiang [13] expanded the TAM and ECM models to examine what factors influence users' continuous intentions in digital museums. Lim, Lim [14] explored work-related conferences and education-related lectures to examine how users' perceived usefulness, perceived convenience and perceived risk influence their intention to use. However, very few studies considered the upside of those virtual meetings and exhibitions. Different from traditional exhibition, virtual exhibition does not generate pollution or cause any environmental issues to the destination. Losada, Jorge [15] have noted that the application of virtual tours has been one of the marketing strategy to encourage sustainable tourism behavior. Recent years, more enterprises emphasize the green culture and corporate environmental responsibility to enhance their brand image [16,17]. It is clear to see the significance of environment-focused contributions for enterprises. For exhibitors, the virtual exhibition can help them to improve the image of sustainability. Thus, researchers must explore whether virtual exhibitions may drive exhibitors' pro-environmental behaviors or not.

In terms of the perceptions of the exhibitions, there are two main streams from the exhibitors' perspective. Many existing studies argue that satisfaction is a key driver to lead customers' and exhibitors' further behavioral intentions. For instance, Chien and Chi [18] explored the relationship among service quality, corporate image, customer satisfaction, and loyalty behavioral intentions by using PLS-SEM. Lee, Lee [19] collected data from 11 hospitality programs and examined what attributes influence students' overall satisfaction in the career expo. Another mainstream argued that perceived performance is widely used among exhibitors in the MICE industry, as it represents the overall judgment of people toward the performance of exhibitions [20,21]. As satisfaction and perceived performance are both crucial mediators affecting behavioral intentions in the exhibition field, it is valuable to consider their mutual interactions and mechanisms in the context of the MICE industry. There are not yet studies considering the effects of both satisfaction and perceived performance on exhibitors' behavioral intentions in the context of the MICE. Therefore, this study influences both these two crucial factors to examine how they work together to influence exhibitors' pro-environmental behaviors.

This study aims to develop the attributes of virtual exhibitions and examine the relationship between the virtual exhibition attributes, perceived performance, satisfaction, and pro-environmental behavior toward exhibitions. The potential contributions of this study are threefold. First, it is a pioneer study to address the relationship between virtual exhibition and pro-environmental behaviors from the exhibitor's perspective. It contributes to MICE industry research by exploring how virtual technologies in exhibition help reduce environmental pollution. Second, the study identifies the overall attributes of virtual exhibition that facilitates researchers to conduct other studies in the context of virtual exhibitions or virtual meetings. Lastly, these results of the study may provide insightful information to exhibition organizers

or virtual technology developers to improve the technologies and applications of virtual exhibitions.

## 2. Literature review

### 2.1 Virtual exhibition

Although the advancement of virtual technologies has been widely applied in the business market, it is still a new concept for many conventional exhibitor enterprises. Many exhibitors gain benefits from traditional offline exhibition, so they do not really care about environmental issues. In recent years, more consumers and enterprises all start considering environmental protection issues [22]. To respond to this, Sisodia, Wolfe [23] mentioned that virtual platform can become a place for stakeholders to contribute collectively with the purpose of environmental protection. From the past literature, it can be seen the importance of virtual exhibition on environmental protection [24,25]. Some exhibitions provide virtual exhibitions and emphasize environmental responsibilities, such as the International Trade Fair on Environmental Protection (Eco Expo Asia) [26]. These enterprises have applied the principles of sustainability practices in their daily operation to protect the earth. To reduce the negative impact on the environment, they always emphasize the environmental-friendly brand culture. Therefore, the virtual exhibition provides a safe and convenient platform service to promote its sustainable development to reduce environmental issues.

In the existing exhibition literature, many scholars have identified the exhibition attributes that can encourage the exhibitors' intention to participate while others emphasize the environmental issues in the exhibition through the literature of eco-development. The question then is how to achieve the sustainable development of the exhibition industry. Many studies in the hospitality field have discussed the use of virtual reality in promoting sustainable development, however, whether the application of virtual tours on exhibition can be useful to support sustainable development is unknown. Therefore, this study reviews the literature to create a universal description of virtual exhibition attributes, which are described as follows.

**2.1.1 Virtual exhibition attributes.** First, a virtual exhibition is defined as "a web-based platform where customers, suppliers, and distributors can get virtually at any time and space" [27]. It presents the convenience for geographically potential buyers [28]. The virtual exhibition usually opens 24 hours and lasts a longer time than the traditional exhibition. In this context, the virtual exhibition shows its characteristics of ubiquity, without the limitation of time and space.

Second, the virtual exhibition focuses on online interactivity that could communicate with participants [27]. In an exhibition, the engagement shows a strong effect on the overall positive experience. Visitors may ask for information about the products and hope to receive on-time feedback from customer service. According to Geigenmüller [27], virtual exhibition enhances continuous communication across time zones. Thus, the virtual platform of an exhibition could be significantly attractive with a higher level of interactivity.

Third, exhibition provides the significant opportunities for information exchange and learning. These are considered the intangible benefits of personal growth [29]. Through virtual exhibitions, online visitors are still concerned about information presentation. It is important that whether they can read rich product information through the virtual platforms. Therefore, information transfer is considered as an important dimension of virtual exhibition.

Fourth, the virtual platform emphasizes the navigation that directs and informs visitors how they could move to the next spatial entity [30]. The digital platform with good navigation in a museum exhibition improves the perceived convenience of access [31]. In the relevant literature of virtual tours, several scholars have identified the importance of navigation on

constructing a satisfactory experience. Accordingly, the dimension of navigation shows the impactful effect on virtual exhibition.

Last, the visual appeal presents increasing attention from visitors. Wu, Jiang (31) noted that the esthetic design of virtual exhibition creates a good atmosphere and feeling for browsing. The visual effects of the exhibition determine whether people choose to visit the virtual exhibition or not [32]. Aesthetics of the website creates a positive feeling toward the products [32], because it stimulates visitors' sensory and helps to generate an immersive experience.

In summary, the past literature shows the attributes of virtual exhibitions mainly focus on five dimensions, which are ubiquity, interactivity, information present, navigation, and visualization. By incorporating these five-dimensional attributes, this research argues that the virtual exhibition with these five attributes will be successful.

## 2.2 Perceived green performance

In the exhibition industry, exhibitors' perceived performance refers to the subjective evaluation of an exhibit's effectiveness, quality, and overall experience by attendees, exhibitors, and other stakeholders [20]. In the past literature, perceived performance, as a perception, is influenced by various factors that contribute to the success and appeal of an exhibition [33]. Optimizing perceived performance can lead to higher satisfaction, increased attendance, and a positive reputation for the event and its organizers. With the focus on environmental protection, many customers tend to move environmental-friendly products, thus, many scholars have started discussing the notion of perceived green performance. Bresciani, Rehman [34] stated that perceived green performance is particularly useful for enterprises who want to achieve environmental consequences of their products. Achieving green performance can help to protect the environment and get a competitive advantage that will invest in the environment [34].

Past literature has proved that the exhibition attributes are the direct factors that influence exhibitors' performance. For example, Whitfield and Webber [35] identified that perceived performance is assessed by the reputation of the exhibition, the presence of exhibitors, and networking opportunities. The characteristics of an exhibition will determine whether exhibitors are satisfied with the overall experience or not. The perceived green performance can be assessed by changes in green technology that is relevant to green development and performance [34]. Therefore, the use of virtual technologies in exhibition industry will largely benefit the environment and help enterprises improve their evaluation of the exhibits. Accordingly, we propose that the attributes of virtual technologies in the exhibition will influence exhibitors' evaluation of the exhibition experience. The following hypotheses are present:

H1a: Navigation positively influences perceived green performance

H1b: Ubiquity positively influences perceived green performance

H1c: Visualization positively influences perceived green performance

H1d: Interactivity positively influences perceived green performance

H1e: Vividness positively influences perceived green performance

## 2.3 Green satisfaction

Improving exhibitors' satisfaction is widely recognized as an important factor leading to the success in MICE industry [36]. Within the highly competitive market, more and more

enterprises take the environment and sustainability as priority that related to their company's missions and culture. As Talwar, Kaur [3] stated: "the virtual technologies can help to reduce environmental issues". In the MICE, many virtual technology companies and exhibition organizers cooperate to organize online exhibition activities. Accordingly, the higher level of satisfaction with the virtual exhibition for exhibitors is important to stand out among other exhibition organizers. Overall, the term "satisfaction" is defined as "the general feeling of pleasure to satisfy customers' expectations, desires, and needs". In the context of present study, following Martínez [37]'s study, "green satisfaction", as an affective feeling, is described as "a pleasure level of virtual experience to satisfy exhibitors' environmental desires and sustainable expectations".

As previous studies suggest, satisfaction and product attributes are positively related [38], because the attributes of a product or service will enhance consumers' satisfaction level. In MICE industry, exhibition presents different format of attributes, thus, generating different level of satisfaction for exhibitors. The extant research has provided evidence that the exhibition attributes positively influence exhibitors' satisfaction. However, in the context of virtual exhibition, whether the virtual attributes will influence exhibitors' environmental-related satisfaction is unknown. Thus, we propose that the virtual attributes that benefit the environment will improve exhibitors' satisfaction. The following hypotheses are present:

H2a: Navigation positively influences green satisfaction

H2b: Ubiquity positively influences green satisfaction

H2c: Visualization positively influences green satisfaction

H2d: Interactivity positively influences green satisfaction

H2e: Vividness positively influences green satisfaction

As Burton, Sheather [39] discussed, consumer satisfaction can be determined through their perceived performance. The relationship between satisfaction and perceived performance has been confirmed in previous studies. In the context of green exhibitions, when exhibitors perceive their overall performance through virtual technologies as positive, they hold a more positive evaluation of satisfaction with virtual exhibition. Accordingly, it can be identified that exhibitors perceived green performance is strongly established to be related with their satisfaction. Thus, the following hypothesis is proposed.

H3: Perceived green performance positively influences green satisfaction

## 2.4 Pro-environmental behavior of continuous virtual exhibition

The pro-environmental behavior indicates people's willingness to reduce environmental impact by several protection actions [40]. Usually, both cognitive and affective aspects present pro-environmental behaviors [41]. The past literature has emphasized the significance of pro-environmental behavior in virtual technologies [3]. Also, some scholars have linked the cognitive and affective factors, such as value and attitude, in examining pro-environmental behaviors in hospitality services [42]. In the current context, the factors that explain pro-environmental behavior consist of perceived green performance and satisfaction. Past scholars have discussed that exhibitors' perceived performance positively influences their future behavioral intention [21]. However, with the application of virtual technologies, no studies have examined the relationship between perceived performance and their future participation in the exhibition. Accordingly, this study proposes the following hypothesis:

H4: Perceived green performance positively influences pro-environmental behavior of continuous virtual exhibition

The extant research has recognized that satisfaction has been identified as a prerequisite to developing behavioral intention. It is generally accepted that exhibitors tend to develop a higher level of intention of future participation in an exhibition when they are satisfied with the experience in an exhibition [43]. In the hospitality and tourism industry, the literature has discussed the importance of consumers' green satisfaction with hotel consumption in developing their future behavioral intentions. In the current background, it is unknown whether exhibitors' satisfaction influences their future willingness to protect the environment by continuous participating in the virtual exhibition. Thus, this study proposes the following hypothesis:

H5: Green satisfaction positively influences pro-environmental behavior of continuous virtual exhibition

Fig 1 presents a proposed model of this study. It consists with attributes level, perception level (cognitive and affective), and behavior level.

## 3. Methodology

### 3.1 Participant and procedure

The research site of this study was targeted to 2021 Zhejiang Yiwu International Intelligent Manufacturing Equipment Expo (ME-Expo), which was organized by the China Chamber of

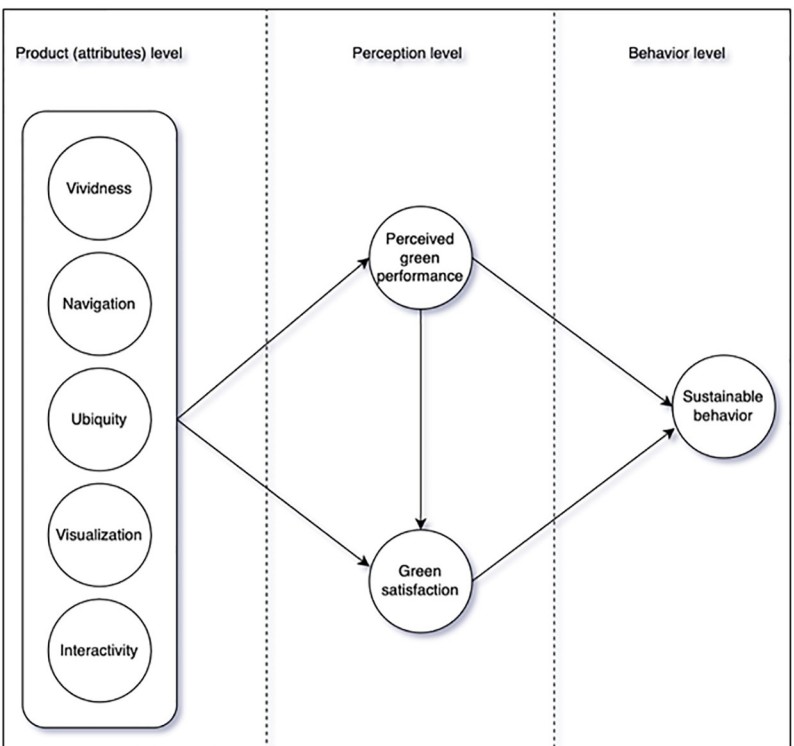

**Fig 1. Proposed model.**

Commerce for Import and Export of Machinery and Electronic and held between March 15[th] to 17[th] in 2022. With the theme of "smart, green and innovative", ME-Expo 2021 set up three major themed areas for the virtual exhibition: plastic packaging area, intelligent warehousing and logistics area, CNC machine tool and metal plastic packaging area, metal processing exhibition area, digital direct injection and personalized customization exhibition area, knitting, sewing equipment, and technology exhibition area. More than 460 companies from 13 provinces of China, the United States, Germany, Japan, Singapore, and Italy participated in the 2021 ME-Expo. The rationale for choosing 2021 ME-Expo is it was mainly promoted by its "innovation and green" theme, which is exactly appropriate in the context of this study. In recent years, more enterprises and exhibition organizers have adopted virtual exhibition to emphasize their green culture. The following 2023 ME-Expo will be still held through the virtual exhibition, with the purpose of green development. The participants were reached by the social network of the 2021 ME-Expo and asked whether they were willing to participate in this research.

In the quantitative section, the participants were invited to fill out the questionnaire online by convenience sampling method, during the 15[th] and 20[th] of 2022. After a critical literature review, this study investigated 8 variables from related literature. The researcher translated all measurement items from English to Chinese and these were checked by 2 English experts with a back-translation method. Three well-trained research assistants collected exhibitors' information and contacted them online to collect data. All participants were told that filling out the questionnaire is voluntary and invited them to finish the electronic informed consent. The screen question was included in the beginning of the questionnaire: "Do you know the main theme of this virtual exhibition is "green" that focuses on environmental protection?" In the beginning, 50 pre-test questionnaires were collected, and the participants did not indicate any problem of understanding items. Finally, a total of 430 questionnaires were collected. Among these, 417 valid questionnaires were applied for further analysis with a 97% response rate. The participants' backgrounds are shown in Table 1. The quantitative and qualitative triangulation methods help ensure a more comprehensive and accurate understanding of the research topic or phenomenon being studied. By using the triangulation method, researchers can mitigate the limitations and biases inherent in quantitative results while increasing the validity and reliability of their findings [44]. This method also allows for a richer interpretation of the research results, enabling a more robust understanding of the phenomenon under investigation [44]. According to previous studies, Cao, Liu [45] and Wong and Pan [46] have also applied the triangulation method to verify the results of the quantitative study. Therefore, this study conducted another qualitative study to verify the reliability of the results.

**Table 1. Participants' background (n = 417).**

| | | Frequency | Percent (%) |
|---|---|---|---|
| Gender | Male | 188 | 45.1 |
| | Female | 229 | 54.9 |
| Company size | 1–50 | 30 | 7.2 |
| | 51–100 | 183 | 43.9 |
| | 101–150 | 83 | 19.9 |
| | 151–200 | 61 | 14.6 |
| | Over 200 | 60 | 14.4 |
| Past experience of online exhibition | First time | 96 | 23.0 |
| | Second time | 90 | 21.6 |
| | Three times and above | 231 | 55.4 |

Table 2. Interviewees' background (n = 18).

| No. | Gender | Occupation | Company size |
|---|---|---|---|
| 1 | Female | Senior manager in food machinery company | 51–100 |
| 2 | Male | Owner in smart storage and logistic company | 1–50 |
| 3 | Male | Assistant marketing manager in plastic and packing machinery company | 1–50 |
| 4 | Female | Customer manager in knitting equipment company | 51–100 |
| 5 | Male | Designer in food machinery company | 1–50 |
| 6 | Male | Seller in mental processing machine company | 1–50 |
| 7 | Male | Assistant manager in high-tech company | 1–50 |
| 8 | Female | Director in Plastic and Packing Machinery company | 51–100 |
| 9 | Female | Equipment developer in smart storage and logistic company | 51–100 |
| 10 | Male | Production operator in equipment development company | 101–150 |
| 11 | Female | Business development manager in new material company | 1–50 |
| 12 | Female | Sales representatives in sewing equipment company | 51–100 |
| 13 | Female | Assistant manager in vacuum cleaners motors company | 101–150 |
| 14 | Male | Director of operation in high-tech company | Over 200 |
| 15 | Male | Sales representatives in machine tool development company | 1–50 |
| 16 | Female | Sourcing manager in high-tech company | 51–100 |
| 17 | Female | Product analyst in food machinery company | 1–50 |
| 18 | Male | Assistant manager in high-tech company | 101–150 |

In qualitative section, 18 organizers and senior exhibitors who did not participate in previous questionnaires were invited for the follow-up interviews. All of them were aged 28–45 and had work experience in the exhibition industry for at least 6 years. At the beginning of the interviews, researcher ensured all of them agreed to participate in this interview and told them all information will be kept confidential. The consent forms were also provided to read and sign. The researcher asked the interview questions based on the guidelines of proposed hypotheses, such as "What kinds of attributes in the virtual exhibition are attractive? And why?" or "What practices will the virtual exhibition organizers can do to trigger exhibitors' pro-environmental behavior? And how?". The previous quantitative results allow us to identify patterns, correlations, and general trends within the dataset. The following qualitative data is to validate the quantitative results and provide industry insights. The average time for a follow-up interview was about fifty minutes. Table 2 shows the background of the interviewees.

## 3.2 Measurement

The original measurement items are all in English, so we followed Behr [47]'s back translation method to ensure the appropriate translation from English to Chinese. The followings are the details of each variable.

**Visualization.** Visualization was measured with a 4-item scale developed from Cheng and Huang [10]. Participants were asked to evaluate their experience on using online exhibitions, using a seven-point Likert scale (1 = strongly disagree, 7 = strongly agree). The sample item is "The view of the museum in the VR tour is harmonious". The Cronbach's alpha for visualization was 0.911.

**Ubiquity.** Ubiquity was measured with a 3-item scale developed from Cheng and Huang [10]. Participants were asked to evaluate their experience on using online exhibition, using a seven-point Likert scale (1 = strongly disagree, 7 = strongly agree). The sample item is "I can join a virtual tour at any time.". The Cronbach's alpha for ubiquity was 0.870.

**Navigation.** Navigation was measured with a 4-item scale developed from Cheng and Huang [10]. Participants were asked to evaluate their experience on using online exhibition, using a seven-point Likert scale (1 = strongly disagree, 7 = strongly agree). The sample item is "I was in control of my movement through the virtual environment (website)". The Cronbach's alpha for navigation was 0.876.

**Vividness.** Vividness was measured with a 3-item scale developed from Kim and Ko [48]. Participants were asked to evaluate their experience of using online exhibitions, using a seven-point Likert scale (1 = strongly disagree, 7 = strongly agree). The sample item is "I thought the sensory information provided by the screen was highly vivid.". The Cronbach's alpha for vividness was 0.894.

**Interactivity.** Interactivity was measured with a 4-item scale developed from Zhou, Li [49]. Participants were asked to evaluate their experience on using online exhibition, using a seven-point Likert scale (1 = strongly disagree, 7 = strongly agree). The sample item is "The virtual platform is effective in gathering consumers' feedback". The Cronbach's alpha for interactivity was 0.917.

**Perceived green performance.** Perceived green performance was measured with a 3-item scale developed from Daily, Bishop [50]. Participants were asked to evaluate their experience on using online exhibition, using a seven-point Likert scale (1 = strongly disagree, 7 = strongly agree). The sample item is "The virtual exhibition efforts have significantly reduced waste within the production process." The Cronbach's alpha for perceived green performance was 0.911.

**Green satisfaction.** Green satisfaction was measured with a 3-item scale developed from Martínez [37]. Participants were asked to evaluate their experience on using online exhibition, using a seven-point Likert scale (1 = strongly disagree, 7 = strongly agree). The sample item is "The choice of this virtual exhibition due to its environmental commitment makes me happy". The Cronbach's alpha for green satisfaction 0.880.

**Sustainable behavior.** Sustainable behavior was measured with a 4-item scale developed from Qin and Luo [8]. Participants were asked to evaluate their experience on using online exhibition, using a seven-point Likert scale (1 = strongly disagree, 7 = strongly agree). The sample item is "Our corporation is willing to adopt eco-exhibiting". The Cronbach's alpha for sustainable behavior 0.885.

## 4. Results

### 4.1 Preliminary analysis

Table 3 presents the descriptive analysis of mean, standard deviation, skewness, and kurtosis. The indicators of excess kurtosis and skewness are greater than -3.0 and less than 3.0, so data are relative to the normal distribution. Further, to evaluate the measurement model, the confirmatory factor analysis was conducted. As shown in Table 3, the factor loadings are all greater than 0.7, suggested by Hair Jr, Sarstedt [51]. As shown in Table 4, the values of Cronbach's alpha and composite reliability (CR) are all higher than the suggested level of 0.7. Also, and the values of the average variance extracted (AVE) are all greater than 0.5, as their recommended level [51]. Accordingly, this study confirms that no issues about convergent validity and internal consistency of the measurement variables. In Table 4, the discriminate validity was also confirmed because the square root of the AVE of each factor exceeds the correlations between potential variables [52] and all values of the Heterotrait-Monotrait ratio (HTMT) were lower than 0.90 [53]. Therefore, the reliability and validity of this study are all confirmed.

**Table 3. Descriptive results of measurement scales.**

| Construct and scale items | FL | SD | Skewness | Kurtosis |
|---|---|---|---|---|
| **Navigation** Cheng and Huang [10] | | | | |
| I was in control of my movement through the virtual environment (website). | 0.850 | 1.114 | -0.972 | 1.440 |
| I had some control over the landscapes I wanted to see in the virtual experience. | 0.862 | 1.232 | -0.546 | -0.062 |
| I was in control over my location. | 0.876 | 1.351 | -0.789 | 0.592 |
| The virtual tourism technology responded to my needs quickly and efficiently. | 0.826 | 1.330 | -0.838 | 0.774 |
| **Ubiquity** Cheng and Huang [10] | | | | |
| I can join a virtual tour at any time. | 0.873 | 1.142 | -0.711 | 0.211 |
| I can join a virtual tour from anywhere. | 0.898 | 1.123 | -0.793 | 0.921 |
| I can participate in a virtual tour when I need to | 0.900 | 1.167 | -0.731 | 0.286 |
| **Vividness** Kim and Ko [48] | | | | |
| I thought the sensory information provided by the screen was highly vivid. | 0.912 | 1.118 | -0.731 | 0.980 |
| I thought the sensory information provided by the screen was highly rich. | 0.925 | 1.157 | -0.926 | 1.345 |
| I thought the sensory contents provided by the screen was highly detailed. | 0.887 | 1.183 | -0.683 | 0.417 |
| **Visualization** Cheng and Huang [10] | | | | |
| The view of the museum in the VR tour is harmonious | 0.899 | 1.456 | -0.626 | 0.088 |
| The museum environment as seen through the VR tour is quite attractive | 0.882 | 1.369 | -0.786 | 0.592 |
| The museum as seen in the VR tour is quite visually appealing. | 0.877 | 1.400 | -0.539 | -0.064 |
| The museum view as seen through the VR tour provided a way for users to easily experience it. | 0.895 | 1.441 | -0.569 | -0.086 |
| **Interactivity** Zhou, Li [49] | | | | |
| The virtual platform is effective in gathering consumers' feedback. | 0.877 | 1.170 | -0.874 | 0.933 |
| The virtual platform gives consumers the opportunity to talk back. | 0.894 | 1.223 | -0.814 | 0.367 |
| The virtual platform facilitates interactive communication between consumers and knowledge contributors. | 0.909 | 1.193 | -0.753 | 0.390 |
| The virtual platform facilitates interactive communication among consumers. | 0.887 | 1.234 | -0.721 | 0.370 |
| **Green satisfaction** Martínez [37]. | | | | |
| The choice of this virtual exhibition due to its environmental commitment makes me happy | 0.883 | 1.472 | -0.594 | 0.060 |
| I consider it is correct to stay in this virtual exhibition because of its environmental commitment | 0.899 | 1.448 | -0.571 | 0.019 |
| I am satisfied with this virtual exhibition because of its environmental performance | 0.912 | 1.481 | -0.562 | -0.007 |
| **Perceived green performance** Daily, Bishop [50] | | | | |
| The virtual exhibition efforts have significantly reduced waste within the production process. | 0.878 | 1.216 | -0.927 | 1.019 |
| The virtual exhibition efforts have significantly improved product quality. | 0.879 | 1.204 | -0.903 | 1.002 |
| Focusing on virtual exhibition has enhanced our facility's reputation. | 0.897 | 1.182 | -0.804 | 0.644 |
| The virtual exhibition efforts have led to improved facility performance. | 0.901 | 1.165 | -0.695 | 0.283 |
| **Sustainable behavior** Qin and Luo [8] | | | | |
| Our corporation is willing to adopt eco-exhibiting. | 0.902 | 1.095 | -0.762 | 0.914 |
| Corporations similar to ours are willing to adopt eco-exhibiting | 0.907 | 1.073 | -0.865 | 1.345 |
| Our corporation is willing to adopt eco-exhibiting during the next 3 years | 0.896 | 1.031 | -0.998 | 2.010 |

Note: FL = factor loading; SD = standard deviation.

## 4.2 Hypotheses testing

The results of hypotheses testing are shown in Table 5. It is significant to test the causal relationships among all variables. The results show five dimensions of online exhibition attributes are all significant related to perceived green performance (H1a: ß = 0.231, t = 8.362; H1b: ß = 0.208, t = 6.552; H1c: ß = 0.151, t = 3.595; H1d: ß = 0.315, t = 6.989; H1e: ß = 0.145, t = 3.954); all five dimensions of online exhibition attributes are all positively related to green satisfaction (H2a: ß = 0.134, t = 3.484; H2b: ß = 0.170, t = 6.307; H2c: ß = 0.128, t = 4.572; H2d: ß = 0.190,

**Table 4. Reliability and validity.**

| | CR | AVE | α | Correlations and the square roots of the AVE* | | | | | | | | HTMT | | | | | | |
|---|---|---|---|---|---|---|---|---|---|---|---|---|---|---|---|---|---|---|
| | | | | 1 | 2 | 3 | 4 | 5 | 6 | 7 | 8 | 1 | 2 | 3 | 4 | 5 | 6 | 7 |
| INTER | 0.941 | 0.800 | 0.917 | **0.894** | | | | | | | | | | | | | | |
| NAV | 0.915 | 0.729 | 0.876 | 0.329 | **0.854** | | | | | | | 0.366 | | | | | | |
| PCP | 0.938 | 0.790 | 0.911 | 0.43 | 0.47 | **0.889** | | | | | | 0.471 | 0.525 | | | | | |
| SA | 0.926 | 0.807 | 0.880 | 0.463 | 0.508 | 0.694 | **0.898** | | | | | 0.515 | 0.577 | 0.774 | | | | |
| BI | 0.929 | 0.813 | 0.885 | 0.462 | 0.547 | 0.655 | 0.691 | **0.902** | | | | 0.51 | 0.619 | 0.725 | 0.78 | | | |
| UBI | 0.920 | 0.793 | 0.870 | 0.322 | 0.286 | 0.427 | 0.461 | 0.477 | **0.891** | | | 0.36 | 0.328 | 0.478 | 0.527 | 0.541 | | |
| VSU | 0.938 | 0.790 | 0.911 | 0.309 | 0.313 | 0.543 | 0.546 | 0.575 | 0.312 | **0.889** | | 0.337 | 0.347 | 0.594 | 0.609 | 0.638 | 0.35 | |
| VID | 0.934 | 0.825 | 0.894 | 0.307 | 0.314 | 0.484 | 0.486 | 0.536 | 0.306 | 0.307 | **0.908** | 0.337 | 0.355 | 0.535 | 0.546 | 0.601 | 0.346 | 0.34 |

Note: INTER = interactivity; NAV = navigation; PCP = perceived green performance; SA = satisfaction; BI = sustainable behavior; UBI = ubiquity; VUS = visualization; VID = vividness.

t = 3.815; H2e: ß = 0.120, t = 3.906); perceived green performance is found to be positively related to both green satisfaction and sustainable behavior (H3: ß = 0.339, t = 4.968; H4: ß = 0.338, t = 4.410); green satisfaction is found positively related to sustainable behavior (H3: ß = 0.456, t = 6.003). The results show all hypotheses are supported.

Furthermore, the values of R-square, showing the prediction of endogenous variables, are all greater than 0.26 [54], as shown in Fig 2. The R-square of perceived green performance is 0.507, 0.605 for satisfaction, 0.537 for sustainable behavior, which all show the sufficient power of the model.

## 4.3 Follow-up interviews

The survey results confirmed that all five attributes significantly affect perceived green performance, green satisfaction, and further influence exhibitors' sustainable behavior. The follow-up interviews were then conducted to gain deeper insights from the senior exhibitors' perspectives. In terms of perceived green performance, the follow-up interviewers confirmed that visualization and vividness are two most important factors leading to perceived green performance, as those two factors are the key to success when promoting green concepts in virtual

**Table 5. Results of proposed hypotheses.**

| | | Path coefficient | T-statistics | VIF | Status |
|---|---|---|---|---|---|
| H1a | Vividness→ Perceived green performance | 0.231 | 8.362 | 1.245 | Support |
| H1b | Navigation→ Perceived green performance | 0.208 | 6.552 | 1.252 | Support |
| H1c | Ubiquity→ Perceived green performance | 0.151 | 3.595 | 1.242 | Support |
| H1d | Visualization→ Perceived green performance | 0.315 | 6.989 | 1.249 | Support |
| H1e | Interactivity→ Perceived green performance | 0.145 | 3.954 | 1.265 | Support |
| H2a | Vividness → Green satisfaction | 0.134 | 3.484 | 1.353 | Support |
| H2b | Navigation → Green satisfaction | 0.170 | 6.307 | 1.340 | Support |
| H2c | Ubiquity → Green satisfaction | 0.128 | 4.572 | 1.289 | Support |
| H2d | Visualization → Green satisfaction | 0.190 | 3.815 | 1.450 | Support |
| H2e | Interactivity → Green satisfaction | 0.120 | 3.906 | 1.308 | Support |
| H3 | Perceived green performance → Green satisfaction | 0.339 | 4.968 | 2.030 | Support |
| H4 | Perceived green performance →Sustainable behavior | 0.338 | 4.410 | 1.928 | Support |
| H5 | Green satisfaction → Sustainable behavior | 0.456 | 6.003 | 1.928 | Support |

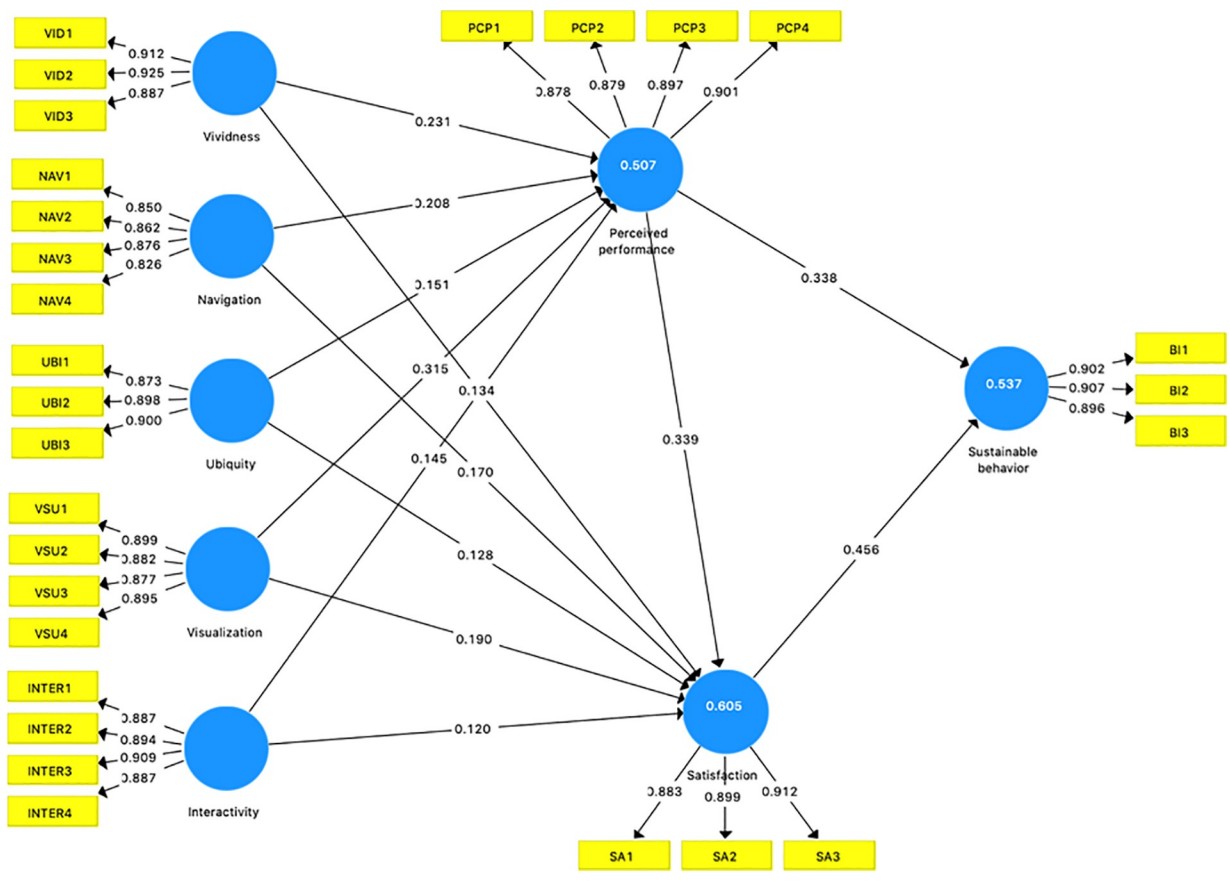

**Fig 2. PLS analysis.**

exhibitions. As one exhibitor said "*Customers may not be able to consider reducing waste or protecting environments at the first glance. They are realistic and therefore the quality of the virtual exhibition is the key before they recognize the green performance.*" Another exhibitor also stated that "*If the visualization is attractive and the information is vivid, they'll feel the virtual exhibition is valuable and therefore believe they may help improve facility performance and quality*"

In terms of green satisfaction, besides visualization and vividness, the interview results also indicate that navigation also plays an important role in generating satisfaction. One exhibitor explains that "*In face-to-face exhibition, customers enjoying their freedom to walk and talk to people freely. However, in virtual exhibition, some exhibitors and customers may find it difficult to control their location, walking speed and so on. So, the movement and the response speed of the virtual exhibition will definitely affect our satisfaction. . .*"

Lastly, to gain sustainable behaviors, exhibitors reflected that both perceived green performance and green satisfaction are important factors in making such a decision, and it is also consistent with the quantitative results. One exhibitor agreed the role of perceived green performance and said "*Surely, if the organizers have not enough capability to perform well in virtual exhibition, they will still highly rely on physical exhibitions, and eco-exhibiting will not be sustainable as it's waste of resources for us (exhibitors) to do both every time. . .*". Another exhibitor explained that why he thinks satisfaction is sometimes even more crucial than perceived green

performance "*Adopting eco-exhibiting is sometimes emotional and not a very objective decision. It means that if we do not feel satisfied with the virtual exhibition, even though we recognize their performance, we will still hesitate and are not willing to use it later...*"

In conclusion, the qualitative results from follow-up interviews supported the proposed model and the results from the quantitative survey.

## 5. Discussion

The present study discusses the importance of virtual technology and its effects on reducing environmental issues. The purpose of this study is to investigate multiple attributes that are involved in virtual exhibition, and their effects on formulating exhibitors' pro-environmental behaviors. The results present five dimensions of examining virtual exhibition attributes: vividness, navigation, ubiquity, visualization, and interactivity. These five dimensions all show the positive relationship with perceived performance and satisfaction. For perceived performance, visualization shows the strongest effect, followed by vividness, navigation, ubiquity, and interactivity. This result confirms Pastel, Chen [55]'s study that emphasize the importance of body visualization on sport performance. For satisfaction, the most important antecedent is visualization, followed by navigation, vividness, ubiquity, and interactivity. This result confirms Berger, Dittenbach [56]'s study that emphasizes the first impression of a website plays the important role for tourism products. While past studies have not yet identified the specific attributes for virtual exhibition, the present study introduced these five dimensions for future exhibition studies and noted that reducing environmental problems should relate to virtual technologies. From the path analysis, the interesting finding is visualization shows the strongest effects on both perceived performance and satisfaction, while interactivity shows the weakest effects on both perceived performance and satisfaction. The virtual technology companies and exhibition organizers will better understand what they should consider as the most important. Although previous studies in social science have broadly explored the use of virtual technologies, there is no study investigating the essence of virtual technologies on environmental protection. This study integrates the different dimensions of virtual exhibition, showing its relevance to develop exhibitors' intention of further pro-environmental behavior.

Additionally, this study helps to understand the mechanism through which satisfaction and perceived performance influence pro-environmental behavior. The results show that satisfaction generates stronger effects on environmental protection behavior, compared with perceived performance. Since perceived performance is related to personal judgment, there is no exact data to support their opinions. To trigger exhibitors' further behavior, satisfaction, as a psychological feeling, becomes the more accurate factor. Therefore, this study shows two different ways for exhibition companies and related stakeholders to generate exhibitors' environmental protection behavior. The first path is composed of their judgment of their product selling generated by virtual attributes, while the second path is composed of emotions and feelings generated by these aspects.

### 5.1 Theoretical implications

The results of this study offer several theoretical implications. First, this study develops a comprehensive set of virtual attributes in exhibitions. Previous studies have identified different attributes of offline exhibition attributes, such as the reputation of the exhibition, access to information, and networking opportunities [35]. By applying virtual technologies, the question of what specific attributes could be applied in the exhibition or MICE industry is unknown. Five dimensions are developed from a critical review of previous research articles, including the literature on virtual technology applications in hospitality and tourism. The identified

virtual exhibition attributes provide a basis for future scholars to examine other attributes with the continuous development of virtual technologies. In more details, the results show that visualization and vividness are the most important attributes in the formation of exhibitors' pro-environmental behavior. Accordingly, the findings provide new insights for the MICE industry that adopts virtual technologies as marketing and environmental conservation strategy.

Second, this study aims to investigate the online exhibition attributes and its effects on exhibitors' sustainable behavior. It fills a research gap in the MICE literature by examining the relationship between virtual technologies in exhibition and environmental protection behavior. The results of the study hope to enhance the theoretical understanding of exhibition development in particular sustainable development. Unlike past studies that have explored several attributes of offline exhibition, this study provides inclusive online attributes to the existing knowledge of exhibition literature. This study provides empirical evidence to explain exhibitors' intention of environmental conservation. The results show that the multi-level virtual technologies do influence exhibitors' perception and behavioral intention of environment protection, thus the relationship between virtual exhibition attributes and environmental protection is confirmed. It is a holistic explanation to the challenges faced in realizing sustainable development. This study extends previous discussions concerning the negative impact of environmental pollution from the exhibition [57], thus contributing to MICE literature.

Third, this study discusses the mutual interaction and mechanism of satisfaction and perceived performance in the context of MICE industry. These two crucial mediators are identified as two main streams influencing pro-environmental behavior. The results show satisfaction and perceived performance all trigger their pro-environmental behavior, in which satisfaction shows stronger effects. The current study provides a better understanding of the determinants of exhibitors' intention to engage in pro-environmental behavior. It also provides a foundation that future scholars may explore other variables that may trigger exhibitors' pro-environmental practices.

Last, this study adopts the triangulation of quantitative questionnaires and qualitative follow-up interviews. The quantitative method helps to explain how the exhibition attributes influence exhibitors' perceptions and behavior scientifically. Following this, the qualitative follow-up interview helps to confirm the importance of exhibition attributes, and their effects on formulating perceived performance, and satisfaction, which, in turn, influence their intention of pro-environmental behavior.

### 5.2 Practical implications

The results of this study provide several practical implications for industry practitioners. First, it provides insightful information for exhibition organizers or virtual technology developers to improve the technologies and applications of virtual exhibitions. More specifically, the results show that visualization is the most important influential factor for both perceived performance and satisfaction. It implies that exhibition organizers should focus on website design. The website should contain multiple elements that are visually appealing and highly attract website viewers. Second, vividness is the second most important attribute of virtual technology. It helps industry practitioners to understand that the sensory information determines exhibitors' interest. After entering the virtual view of the exhibition, whether the product information provided by the screen is detailed and rich becomes another important dimension of attracting exhibitors. Based on the results of this study, to gain visualization, the organizers can incorporate VR technology, offering a more immersive experience to exhibitors and visitors. This can include creating virtual showrooms or product demonstrations that simulate real-life experiences, enhancing the visual impact and clarity. In addition, the organizer can provide high-

resolution images, videos, and interactive content for exhibitors to showcase their products or services. This can be achieved through advanced technology or partnerships with digital content providers. In this order, exhibition organizers and virtual technologies developers can better understand what motivates exhibitors to engage in virtual exhibition activities. Also, the results of this study show the importance of vividness, therefore, the organizer can provide high-resolution images, videos, and interactive content for exhibitors to showcase their products or services. This can be achieved through advanced technology or partnerships with digital content providers.

Our findings also show that satisfaction greatly influences exhibitors' intention of pro-environmental behavior, while perceived performance shows fewer effects. Hence, under the current background that the whole society actively encourages sustainability, triggering exhibitors' pro-environmental behavior will benefit the whole exhibition industry. When determining the factors that influence exhibitors' intention, satisfying exhibitors is considered as the most important strategy. We suggest that the exhibition organizers may consider providing a better comprehensive service and environmental image of the enterprises, to induce their future behavior of continuously adopting the virtual technologies. More specifically, the organizers can encourage participants to attend the trade show virtually instead of physically by highlighting the environmental benefits of reducing travel and carbon emissions. They should also emphasize the convenience and cost-effectiveness of virtual participation. Furthermore, the organizers may create a dedicated section or forum within the virtual trade show platform to promote sustainability practices, such as facilitate discussions, share success stories, and provide resources on how companies can integrate sustainability into their business operations. By implementing these practical strategies, the organizers can enhance the environmental impact of the virtual trade show and further engage participating companies in their social responsibility towards environmental conservation.

### 5.3 Limitations and directions for future research

This study contains several limitations that should be addressed in future research. Firstly, the samples of this study were collected from 417 through the online channel. The results may not have a generalization globally. The future research may consider collecting data in other regions and countries. Second, this study examined exhibitors' intention of pro-environmental behavior. it is unknown what their actual pro-environmental behavior is in the future. Therefore, the future researcher may consider re-examining exhibitors' actual behavior in the future research model. Third, this research applied multi-level exhibition attributes to explain perceived performance and satisfaction. Some unmeasured variables may be explored in future research.

## Supporting information

**S1 Data.**
(CSV)

**S1 File.**
(DOCX)

## Author Contributions

**Conceptualization:** Qing Xia, Jose Weng Chou Wong.

**Formal analysis:** Shan Wang.

**Investigation:** Qing Xia.

**Methodology:** Shan Wang, Jose Weng Chou Wong.

**Project administration:** Qing Xia.

**Validation:** Shan Wang.

**Writing – original draft:** Shan Wang, Jose Weng Chou Wong.

**Writing – review & editing:** Qing Xia.

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
