## [Decision Letter · Decision Letter 0]

2 Oct 2023

PONE-D-23-26178The use of virtual exhibition to promote exhibitors’ pro-environmental behavior: the case study of 2021 Zhejiang Yiwu International Intelligent Manufacturing Equipment ExpoPLOS ONE

Dear Dr. Wang,

Thank you for submitting your manuscript to PLOS ONE. After careful consideration, we feel that it has merit but does not fully meet PLOS ONE’s publication criteria as it currently stands. Therefore, we invite you to submit a revised version of the manuscript that addresses the points raised during the review process.

* Need to present the source of the scale item used in your study. Good if it is presented in Appendix. * Purpose of conducting interview after survey need to be explained clearly in the methodology. 

We look forward to receiving your revised manuscript.

Kind regards,

Sudarsan Jayasingh, Ph.D

Academic Editor

PLOS ONE

Journal Requirements:

Reviewers' comments:

Reviewer's Responses to Questions

**Comments to the Author**

1. Is the manuscript technically sound, and do the data support the conclusions?

Reviewer #1: Yes

Reviewer #2: Yes

2. Has the statistical analysis been performed appropriately and rigorously? 

Reviewer #1: Yes

Reviewer #2: Yes

3. Have the authors made all data underlying the findings in their manuscript fully available?

Reviewer #1: No

Reviewer #2: No

4. Is the manuscript presented in an intelligible fashion and written in standard English?

Reviewer #1: Yes

Reviewer #2: Yes

5. Review Comments to the Author

Reviewer #1: 1.The format of each paragraph needs to be uniform.

2.Advice for the exhibition industry could be added.

3.Please attach empirical data.

4.The meaning can be considered in the first half of the article.

Reviewer #2: Thanks for letting me review your exciting manuscript.

Regarding the form, I propose the necessity of some language-in-use corrections with some English editing.

In terms of the content, Although the statistical analysis in the manuscript is correct, your procedure demands better sustaining the relationship between the content analysis of the interviews and the quantitative PLS statistical analysis. This aspect requires extra space in the methods section, additional details of your analytic approach, and qualitative interviews' data use.

It is also imperative to explain how quantitative data is supported and mixed up with the data from interviews.

All in all, with those additions, the journal must accept your article.

6. PLOS authors have the option to publish the peer review history of their article (what does this mean?). If published, this will include your full peer review and any attached files.

Reviewer #1: No

Reviewer #2: **Yes: **Juan Felipe Espinosa-Cristia, PhD

---

## [Author Response · Author response to Decision Letter 0]

20 Oct 2023

Response to editor:

1. Need to present the source of the scale item used in your study. Good if it is presented in Appendix.

Response: Thanks for editor’s comment. We have already inserted the sources of the scale items used in this study, as Table 3 shows (Page 20, Line 364). In addition, we have attached a new Appendix 1 (Page 38-40) at the end of the manuscript, showing the questionnaire of this study. 

2. Purpose of conducting interview after survey need to be explained clearly in the methodology.

Response: Thanks for editor’s comment. We have added some contents that clearly explained the purpose of conducting the interview after survey in methodology section (Page 15, Line 280-288; Page 16, Line 294-299). The new added sentences are shown below: 

---“The quantitative and qualitative triangulation methods help ensure a more comprehensive and accurate understanding of the research topic or phenomenon being studied. By using the triangulation method, researchers can mitigate the limitations and biases inherent in quantitative results while increasing the validity and reliability of their findings [44]. This method also allows for a richer interpretation of the research results, enabling a more robust understanding of the phenomenon under investigation [44]. According to previous studies, Cao, Liu (45) and Wong and Pan (46) have also applied the triangulation method to verify the results of the quantitative study. Therefore, this study conducted another qualitative study to verify the reliability of the results.” 

--- “The researcher asked the interview questions based on the guideline of proposed hypotheses, such as “what kinds of attributes in the virtual exhibition are attractive? And why?” or “what practices will the virtual exhibition organizers can do to trigger exhibitors’ pro-environmental behavior? And how?”. The previous quantitative results allow us to identify patterns, correlations, and general trends within the dataset. The following qualitative data is to validate the quantitative results and provide industry insights”.

Response to Reviewer #1:

1. The format of each paragraph needs to be uniform.

Response: Thanks for reviewer’s comment. We have checked the format of each paragraph being uniform. 

2.Advice for the exhibition industry could be added.

Response: Thanks for reviewer’s comment. We have added some advices for the exhibition industry at the practical implications section (Page 32, Line 510-519; Page 33, Line 527-535) , such as: 

--- “The organizers can incorporate VR technology, offerring a more immersive experience to exhibitors and visitors. This can include creating virtual showrooms or product demonstrations that simulate real-life experiences, enhancing the visual impact and clarity. In addition, the organizer can provide high-resolution images, videos, and interactive content for exhibitors to showcase their products or services. This can be achieved through advanced technology or partnerships with digital content providers”

--- “The organizer can provide high-resolution images, videos, and interactive content for exhibitors to showcase their products or services. This can be achieved through advanced technology or partnerships with digital content providers

--- “Furthermore, the organizers can create a dedicated section or forum within the virtual trade show platform to promote sustainability practices, such as facilitate discussions, share success stories, and provide resources on how companies can integrate sustainability into their business operations”. 

3.Please attach empirical data. The meaning can be considered in the first half of the article.

Response: Thanks for reviewer’s comment. The empirical data has been attached at Mendeley Data official website. Please refer to DOI:10.17632/b34wj2dvnv.1

The citation of data is: Wang, Shan (2023), “The use of virtual exhibition to promote exhibitors’ pro-environmental behavior: the case study of 2021 Zhejiang Yiwu International Intelligent Manufacturing Equipment Expo”, Mendeley Data, V1, doi: 10.17632/b34wj2dvnv.1

Response to Reviewer #2:

1.Thanks for letting me review your exciting manuscript. Regarding the form, I propose the necessity of some language-in-use corrections with some English editing.

Response: Thanks for reviewer’s comment. We have invited a professional English translator to finish the proofreading of this manuscript. 

2.In terms of the content, Although the statistical analysis in the manuscript is correct, your procedure demands better sustaining the relationship between the content analysis of the interviews and the quantitative PLS statistical analysis. This aspect requires extra space in the methods section, additional details of your analytic approach, and qualitative interviews' data use. 

 Response: Thanks for reviewer’s comment. We have added a paragraph that clearly explained the purpose of conducting the interview after survey in methodology section. The new added sentences are shown below (Page 15, Line 280-288): 

---“ The quantitative and qualitative triangulation methods help ensure a more comprehensive and accurate understanding of the research topic or phenomenon being studied. By using the triangulation method, researchers can mitigate the limitations and biases inherent in quantitative results while increasing the validity and reliability of their findings [44]. This method also allows for a richer interpretation of the research results, enabling a more robust understanding of the phenomenon under investigation [44]. According to previous studies, Cao, Liu (45) and Wong and Pan (46) have also applied the triangulation method to verify the results of the quantitative study. Therefore, this study conducted another qualitative study to verify the reliability of the results.” 

3.It is also imperative to explain how quantitative data is supported and mixed up with the data from interviews. All in all, with those additions, the journal must accept your article.

Response: Thanks for reviewer’s valuable feedback. In our manuscript, we applied the quantitative and qualitative triangulation method to ensure a comprehensive analysis of the research topic. In methodology section, we have explained the significance of conducting interview after the questionnaire. The quantitative data were collected through surveys/questionnaires administered to a large sample size, allowing us to obtain a broad understanding of participants' viewpoints and experiences related to the research topic. After this, the interviewees were firstly asked by following a basic interview guideline related to the topic with questions like “what kinds of attributes in the virtual exhibition are attractive?” or “what practices will the virtual exhibition organizers can do to trigger exhibitors’ pro-environmental behavior?”. The previous quantitative results allow us to identify patterns, correlations, and general trends within the dataset. The following qualitative data is to validate the quantitative results and provide industry insights. After completing individual analyses, we employed a process of data comparison and integration, examining how the qualitative insights either confirmed, expanded, or provided further explanation to the quantitative findings. This process facilitated an iterative triangulation approach that strengthened the validity and reliability of our study. To illustrate this integration in our manuscript, we have revised the methodology section to provide a clear and detailed description of the process involved in supporting and mixing the quantitative and qualitative data (Page 15-16, Line 280-297). We recognize the importance of making this aspect transparent for readers, ensuring they can fully grasp the rigor and comprehensiveness of our analysis. Thank you again for your constructive feedback, which will significantly enhance the clarity and quality of our manuscript.

---

## [Editor Report · Decision Letter 1]

3 Nov 2023

The use of virtual exhibition to promote exhibitors’ pro-environmental behavior: the case study of Zhejiang Yiwu International Intelligent Manufacturing Equipment Expo

PONE-D-23-26178R1

Dear Dr. Wang,

We’re pleased to inform you that your manuscript has been judged scientifically suitable for publication and will be formally accepted for publication once it meets all outstanding technical requirements.

Kind regards,

Sudarsan Jayasingh, Ph.D

Academic Editor

PLOS ONE
---

## [Editor Report · Acceptance letter]

8 Nov 2023

PONE-D-23-26178R1 

The use of virtual exhibition to promote exhibitors’ pro-environmental behavior: the case study of Zhejiang Yiwu International Intelligent Manufacturing Equipment Expo 

Dear Dr. Wang:

I'm pleased to inform you that your manuscript has been deemed suitable for publication in PLOS ONE. Congratulations! Your manuscript is now with our production department. 

Kind regards, 

on behalf of

Dr. Sudarsan Jayasingh 

Academic Editor

PLOS ONE